# Summary Statistics for Partitionings and Feature Allocations

**Işık Barış Fidaner**
Computer Engineering Department
Boğaziçi University, Istanbul
fidaner@alternatifbilisim.org

**Ali Taylan Cemgil**
Computer Engineering Department
Boğaziçi University, Istanbul
taylan.cemgil@boun.edu.tr

## Abstract

Infinite mixture models are commonly used for clustering. One can sample from the posterior of mixture assignments by Monte Carlo methods or find its *maximum a posteriori* solution by optimization. However, in some problems the posterior is diffuse and it is hard to interpret the sampled partitionings. In this paper, we introduce novel statistics based on block sizes for representing sample sets of partitionings and feature allocations. We develop an element-based definition of entropy to quantify segmentation among their elements. Then we propose a simple algorithm called *entropy agglomeration* (EA) to summarize and visualize this information. Experiments on various infinite mixture posteriors as well as a feature allocation dataset demonstrate that the proposed statistics are useful in practice.

## 1   Introduction

Clustering aims to summarize observed data by grouping its elements according to their similarities. Depending on the application, clusters may represent words belonging to topics, genes belonging to metabolic processes or any other relation assumed by the deployed approach. Infinite mixture models provide a general solution by allowing a potentially unlimited number of mixture components. These models are based on nonparametric priors such as Dirichlet process (DP) [1, 2], its superclass Poisson-Dirichlet process (PDP) [3, 4] and constructions such as Chinese restaurant process (CRP) [5] and stick-breaking process [6] that enable formulations of efficient inference methods [7]. Studies on infinite mixture models inspired the development of several other models [8, 9] including Indian buffet process (IBP) for infinite feature models [10, 11] and fragmentation-coagulation process for sequence data [12] all of which belong to Bayesian nonparametrics [13].

In making inference on infinite mixture models, a sample set of partitionings can be obtained from the posterior.[1] If the posterior is peaked around a single partitioning, then the *maximum a posteriori* solution will be quite informative. However, in some cases the posterior is more diffuse and one needs to extract statistical information about the *random partitioning* induced by the model. This problem to 'summarize' the samples from the infinite mixture posterior was raised in bioinformatics literature in 2002 by Medvedovic and Sivaganesan for clustering gene expression profiles [14]. But the question proved difficult and they 'circumvented' it by using a heuristic linkage algorithm based on pairwise occurence probabilities [15, 16]. In this paper, we approach this problem and propose basic methodology for summarizing sample sets of partitionings as well as feature allocations.

Nemenman *et al.* showed in 2002 that the entropy [17] of a DP posterior was strongly determined by its prior hyperparameters [18]. Archer *et al.* recently elaborated these results with respect to PDP [19]. In other work, entropy was generalized to partitionings by interpreting partitionings as probability distributions [20, 21]. Therefore, entropy emerges as an important statistic for our problem, but new definitions will be needed for quantifying information in feature allocations.

In the following sections, we define the problem and introduce *cumulative statistics* for representing partitionings and feature allocations. Then, we develop an interpretation for entropy function in terms of *per-element information* in order to quantify segmentation among their elements. Finally, we describe *entropy agglomeration* (EA) algorithm that generates dendrograms to summarize sample sets of partitionings and feature allocations. We demonstrate EA on infinite mixture posteriors for synthetic and real datasets as well as on a real dataset directly interpreted as a feature allocation.

## 2  Basic definitions and the motivating problem

We begin with basic definitions. A *partitioning* of a set of elements $[n] = \{1, 2, \ldots, n\}$ is a set of blocks $Z = \{B_1, \ldots, B_{|Z|}\}$ such that $B_i \subset [n]$ and $B_i \neq \emptyset$ for all $i \in \{1, \ldots, n\}$, $B_i \cap B_j = \emptyset$ for all $i \neq j$, and $\cup_i B_i = [n]$.[2] We write $Z \vdash [n]$ to designate that $Z$ is a partitioning of $[n]$.[3] A sample set $E = \{Z^{(1)}, \ldots, Z^{(T)}\}$ from a distribution $\pi(Z)$ over partitionings is a multiset such that $Z^{(t)} \sim \pi(Z)$ for all $t \in \{1, \ldots, T\}$. We are required to extract information from this sample set.

Our motivation is the following problem: a set of observed elements $(x_1, \ldots, x_n)$ are clustered by an infinite mixture model with parameters $\theta^{(k)}$ for each component $k$ and mixture assignments $(z_1, \ldots, z_n)$ drawn from a two-parameter CRP prior with concentration $\alpha$ and discount $d$ [5].

$$z \sim CRP(z; \alpha, d) \qquad \theta^{(k)} \sim p(\theta) \qquad x_i \mid z_i, \theta \sim F(x_i \mid \theta^{(z_i)}) \qquad (1)$$

In the conjugate case, all $\theta^{(k)}$ can be integrated out to get $p(z_i \mid z_{-i}, x)$ for sampling $z_i$ [22]:

$$p(z_i \mid z_{-i}, x) \propto \int p(z, x, \theta)\, d\theta \propto \begin{cases} \frac{n_k - d}{n - 1 + \alpha} \int F(x_i|\theta)\, p(\theta|x_{-i}, z_{-i})\, d\theta & \text{if } k \leq K^+ \\ \frac{\alpha + dK^+}{n - 1 + \alpha} \int F(x_i|\theta)\, p(\theta)\, d\theta & \text{otherwise} \end{cases} \qquad (2)$$

There are $K^+$ non-empty components and $n_k$ elements in each component $k$. In each iteration, $x_i$ will either be put into an existing component $k \leq K^+$ or it will be assigned to a new component. By sampling all $z_i$ repeatedly, a sample set of assignments $z^{(t)}$ are obtained from the posterior $p(z \mid x) = \pi(Z)$. These $z^{(t)}$ are then represented by partitionings $Z^{(t)} \vdash [n]$. The induced sample set contains information regarding (1) CRP prior over partitioning structure given by the hyperparameters $(\alpha, d)$ and (2) integrals over $\theta$ that capture the relation among the observed elements $(x_1, \ldots, x_n)$.

In addition, we aim to extract information from feature allocations, which constitute a superclass of partitionings [11]. A *feature allocation* of $[n]$ is a multiset of blocks $F = \{B_1, \ldots, B_{|F|}\}$ such that $B_i \subset [n]$ and $B_i \neq \emptyset$ for all $i \in \{1, \ldots, n\}$. A sample set $E = \{F^{(1)}, \ldots, F^{(T)}\}$ from a distribution $\pi(F)$ over feature allocations is a multiset such that $F^{(t)} \sim \pi(F)$ for all $t$. Current exposition will focus on partitionings, but we are also going to show how our statistics apply to feature allocations.

Assume that we have obtained a sample set $E$ of partitionings. If it was obtained by sampling from an infinite mixture posterior, then its blocks $B \in Z^{(t)}$ correspond to the mixture components. Given a sample set $E$, we can approximate any statistic $f(Z)$ over $\pi(Z)$ by averaging it over the set $E$:

$$Z^{(1)}, \ldots, Z^{(T)} \sim \pi(Z) \qquad \Rightarrow \qquad \frac{1}{T} \sum_{t=1}^{T} f(Z^{(t)}) \approx \langle\, f(Z)\, \rangle_{\pi(Z)} \qquad (3)$$

Which $f(Z)$ would be a useful statistic for $Z$? Three statistics commonly appear in the literature:

First one is the *number of blocks* $|Z|$, which has been analyzed theoretically for various nonparametric priors [2, 5]. It is simple, general and exchangable with respect to the elements $[n]$, but it is not very informative about the distribution $\pi(Z)$ and therefore is not very useful in practice.

A common statistic is *pairwise occurence*, which is used to extract information from infinite mixture posteriors in applications like bioinformatics [14]. For given pairs of elements $\{a, b\}$, it counts the number of blocks that contain these pairs, written $\sum_i [\{a, b\} \subset B_i]$. It is a very useful similarity measure, but it cannot express information regarding relations among three or more elements.

Another statistic is *exact block size distribution* (referred to as 'multiplicities' in [11, 19]). It counts the partitioning's blocks that contain exactly $k$ elements, written $\sum_i [|B_i| = k]$. It is exchangable with respect to the elements $[n]$, but its weighted average over a sample set is difficult to interpret.

Let us illustrate the problem by a practical example, to which we will return in the formulations:

$$E_3 = \{Z^{(1)}, Z^{(2)}, Z^{(3)}\}$$
$$Z^{(1)} = \{\{1,3,6,7\},\{2\},\{4,5\}\} \qquad S_1 = \{1,2,3,4\}$$
$$Z^{(2)} = \{\{1,3,6\},\{2,7\},\{4,5\}\} \qquad S_2 = \{1,3,6,7\}$$
$$Z^{(3)} = \{\{1,2,3,6,7\},\{4,5\}\} \qquad S_3 = \{1,2,3\}$$

Suppose that $E_3$ represents interactions among seven genes. We want to compare the subsets of these genes $S_1$, $S_2$, $S_3$. The *projection* of a partitioning $Z \vdash [n]$ onto $S \subset [n]$ is defined as the set of non-empty intersections between $S$ and $B \in Z$. Projection onto $S$ induces a partitioning of $S$.

$$PROJ(Z,S) = \{B \cap S\}_{B \in Z} \setminus \{\emptyset\} \qquad \Rightarrow \qquad PROJ(Z,S) \vdash S \qquad (4)$$

Let us represent gene interactions in $Z^{(1)}$ and $Z^{(2)}$ by projecting them onto each of the given subsets:

$$PROJ(Z^{(1)}, S_1) = \{\{1,3\},\{2\},\{4\}\} \qquad PROJ(Z^{(2)}, S_1) = \{\{1,3\},\{2\},\{4\}\}$$
$$PROJ(Z^{(1)}, S_2) = \{\{1,3,6,7\}\} \qquad PROJ(Z^{(2)}, S_2) = \{\{1,3,6\},\{7\}\}$$
$$PROJ(Z^{(1)}, S_3) = \{\{1,3\},\{2\}\} \qquad PROJ(Z^{(2)}, S_3) = \{\{1,3\},\{2\}\}$$

Comparing $S_1$ to $S_2$, we can say that $S_1$ is 'more segmented' than $S_2$, and therefore genes in $S_2$ should be more closely related than those in $S_1$. However, it is more subtle and difficult to compare $S_2$ to $S_3$. A clear understanding would allow us to explore the subsets $S \subset [n]$ in an informed manner. In the following section, we develop a novel and general approach based on block sizes that opens up a systematic method for analyzing sample sets over partitionings and feature allocations.

## 3 Cumulative statistics to represent structure

We define *cumulative block size distribution*, or 'cumulative statistic' in short, as the function $\phi_k(Z) = \sum_i [|B_i| \geq k]$, which counts the partitioning's blocks of size at least $k$. We can rewrite the previous statistics: number of blocks as $\phi_1(Z)$, exact block size distribution as $\phi_k(Z) - \phi_{k+1}(Z)$, and pairwise occurence as $\phi_2(PROJ(Z,\{a,b\}))$. Moreover, cumulative statistics satisfy the following property: for partitionings of $[n]$, $\phi(Z)$ always sums up to $n$, just like a probability mass function that sums up to 1. When blocks of $Z$ are sorted according to their sizes and the indicators $[|B_i| \geq k]$ are arranged on a matrix as in Figure 1a, they form a Young diagram, showing that $\phi(Z)$ is always the conjugate partition of the integer partition of $Z$. As a result, $\phi(Z)$ as well as weighted averages over several $\phi(Z)$ always sum up to $n$, just like taking averages over probability mass functions (Figure 2). Therefore, cumulative statistics of a random partitioning 'conserve mass'. In the

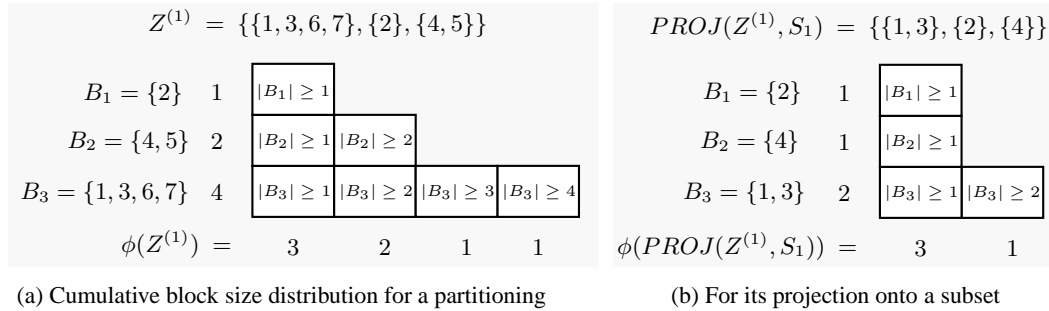

(a) Cumulative block size distribution for a partitioning    (b) For its projection onto a subset

Figure 1: Young diagrams show the conjugacy between a partitioning and its cumulative statistic

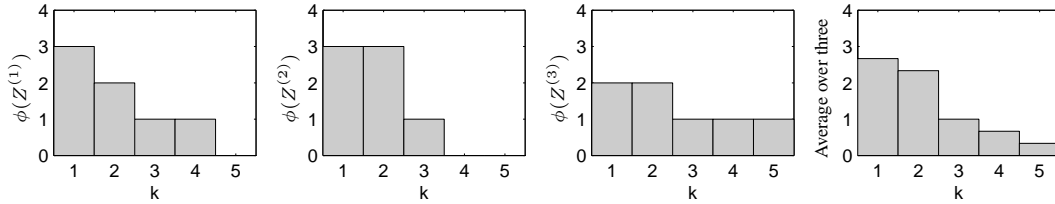

Figure 2: Cumulative statistics of the three examples and their average: all sum up to 7

case of feature allocations, since elements can be omitted or repeated, this property does not hold.

$$Z \vdash [n] \quad \Rightarrow \quad \sum_{k=1}^{n} \phi_k(Z) = n \quad \Rightarrow \quad \sum_{k=1}^{n} \langle\, \phi_k(Z) \,\rangle_{\pi(Z)} = n \qquad (5)$$

When we project the partitioning $Z$ onto a subset $S \subset [n]$, the resulting vector $\phi(PROJ(Z,S))$ will then sum up to $|S|$ (Figure 1b). A 'taller' Young diagram implies a 'more segmented' subset.

We can form a partitioning $Z$ by inserting elements $1, 2, 3, 4, \ldots$ into its blocks (Figure 3a). In such a scheme, each step brings a new element and requires a new decision that will depend on all previous decisions. It would be better if we could determine the whole path by few initial decisions.

Now suppose that we know $Z$ from the start and we generate an incremental sequence of subsets $S_1 = \{1\}$, $S_2 = \{1, 2\}$, $S_3 = \{1, 2, 3\}$, $S_4 = \{1, 2, 3, 4\}$, ... according to a permutation of $[n]$: $\sigma = (1, 2, 3, 4, \ldots)$. We can then represent any path in Figure 3a by a sequence of $PROJ(Z, S_i)$ and determine the whole path by two initial parameters: $Z$ and $\sigma$. The resulting tree can be simplified by representing the partitionings by their cumulative statistics instead of their blocks (Figure 3b).

Based on this concept, we define *cumulative occurence distribution* (COD) as the triangular matrix of incremental cumulative statistic vectors, written $\Delta_{i,k}(Z, \sigma) = \phi_k(PROJ(Z, S_i))$ where $Z \vdash [n]$, $\sigma$ is a permutation of $[n]$ and $S_i = \{\sigma_1, \ldots, \sigma_i\}$ for $i \in \{1, \ldots, n\}$. COD matrices for two extreme paths (Figure 3c, 3e) and for the example partitioning $Z^{(1)}$ (Figure 3d) are shown. For partitionings, $i$th row of a COD matrix always sums up to $i$, even when averaged over a sample set as in Figure 4.

$$Z \vdash [n] \quad \Rightarrow \quad \sum_{k=1}^{i} \Delta_{i,k}(Z, \sigma) = i \quad \Rightarrow \quad \sum_{k=1}^{i} \langle\, \Delta_{i,k}(Z, \sigma) \,\rangle_{\pi(Z)} = i \qquad (6)$$

Expected COD matrix of a random partitioning expresses (1) cumulation of elements by the differences between its rows, and (2) cumulation of block sizes by the differences between its columns.

As an illustrative example, consider $\pi(Z) = CRP(Z|\alpha, d)$. Since CRP is exchangable and projective, its expected cumulative statistic $\langle\phi(Z)\rangle_{\pi(Z)}$ for $n$ elements depends only on its hyperparameters $(\alpha, d)$. As a result, its expected COD matrix $\Delta = \langle\Delta(Z, \sigma)\rangle_{\pi(Z)}$ is independent of $\sigma$, and it

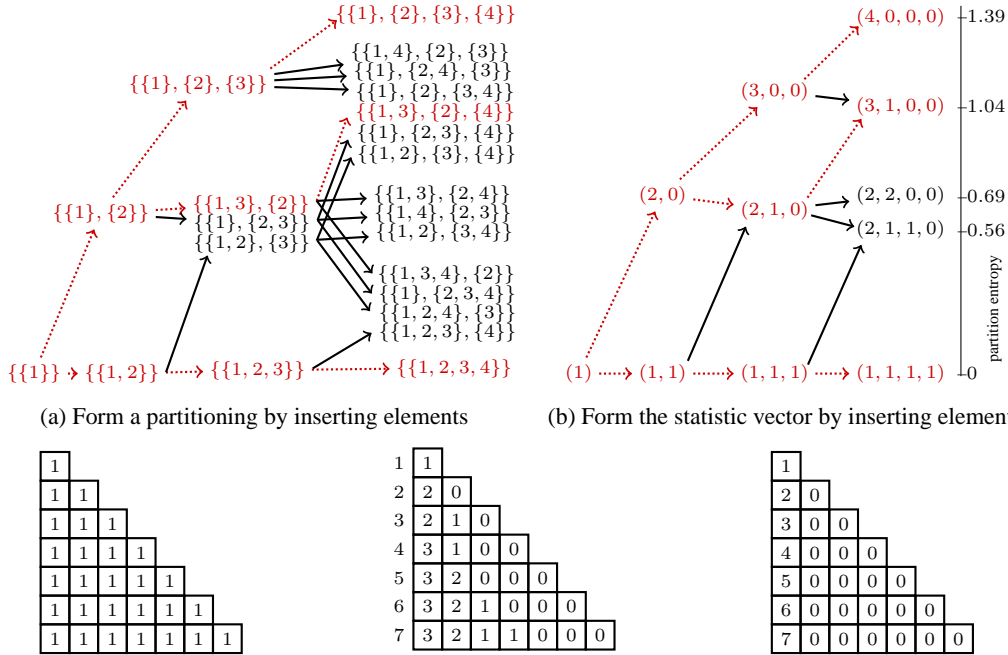

(a) Form a partitioning by inserting elements
(b) Form the statistic vector by inserting elements

(c) All elements into one block  (d) COD matrix $\Delta(Z^{(1)}, (1, \ldots, 7))$  (e) Each element into a new block

Figure 3: Three COD matrices correspond to the three red dotted paths on the trees above

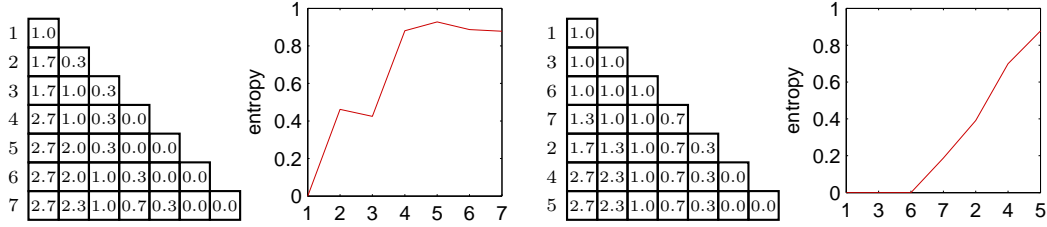

Figure 4: CODs and entropies over $E_3$ for permutations $(1, 2, 3, 4, 5, 6, 7)$ and $(1, 3, 6, 7, 2, 4, 5)$

satisfies an incremental formulation with the parameters $(\alpha, d)$ over the indices $i \in \mathbb{N}$, $k \in \mathbb{Z}^+$:

$$\Delta_{0,k} = 0 \qquad \Delta_{i+1,k} = \Delta_{i,k} + \begin{cases} \frac{\alpha + d\Delta_{i,k}}{i+\alpha} & \text{if } k = 1 \\ \frac{(k-1-d)(\Delta_{i,k-1} - \Delta_{i,k})}{i+\alpha} & \text{otherwise} \end{cases} \tag{7}$$

By allowing $k = 0$ and setting $\Delta_{i,0} = -\frac{\alpha}{d}$, and $\Delta_{0,k} = 0$ for $k > 0$ as the two boundary conditions, the same matrix can be formulated by a difference equation over the indices $i \in \mathbb{N}$, $k \in \mathbb{N}$:

$$(\Delta_{i+1,k} - \Delta_{i,k})(i + \alpha) = (\Delta_{i,k-1} - \Delta_{i,k})(k - 1 - d) \tag{8}$$

By setting $\Delta = \Delta^{(0)}$ we get an infinite sequence of matrices $\Delta^{(m)}$ that satisfy the same equation:

$$(\Delta_{i+1,k}^{(m)} - \Delta_{i,k}^{(m)})(i + \alpha) = (\Delta_{i,k-1}^{(m)} - \Delta_{i,k}^{(m)})(k - 1 - d) = \Delta_{i,k}^{(m+1)} \tag{9}$$

Therefore, expected COD matrix of a CRP-distributed random partitioning is at a constant 'equilibrium' determined by $\alpha$ and $d$. This example shows that the COD matrix can reveal specific information about a distribution over partitionings; of course in practice we encounter non-exchangeable and almost arbitrary distributions over partitionings (e.g., the posterior distribution of an infinite mixture), therefore in the following section we will develop a measure to quantify this information.

## 4   Entropy to quantify segmentation

Shannon's entropy [17] can be an appropriate quantity to measure 'segmentation' with respect to partitionings, which can be interpreted as probability distributions [20, 21]. Since this interpretation does not cover feature allocations, we will make an alternative, element-based definition of entropy.

How does a block $B$ inform us about its elements? Each element has a proportion $1/|B|$, let us call this quantity *per-element segment size*. Information is zero for $|B| = n$, since $1/n$ is the minimum possible segment size. If $|B| < n$, the block supplies positive information since the segment size is larger than minimum, and we know that *its segment size could be smaller if the block were larger*. To quantify this information, we define *per-element information* for a block $B$ as the integral of segment size $1/s$ over the range $[|B|, n]$ of block sizes that make this segment smaller (Figure 5).

$$\mathrm{pei}_n(B) = \int_{|B|}^{n} \frac{1}{s} \, ds = \log \frac{n}{|B|} \tag{10}$$

In $\mathrm{pei}_n(B)$, $n$ is a 'base' that determines the minimum possible per-element segment size. Since segment size expresses the *significance* of elements, the function integrates segment sizes over the block sizes that make the elements *less significant*. This definition is comparable to the well-known *p-value*, which integrates probabilities over the values that make the observations *more significant*.

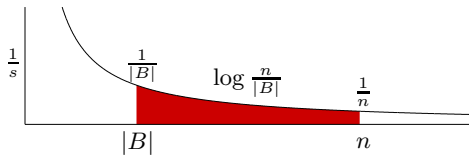

Figure 5: Per-element information for $B$

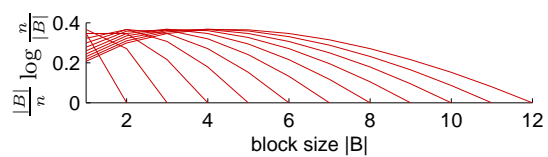

Figure 6: Weighted information plotted for each $n$

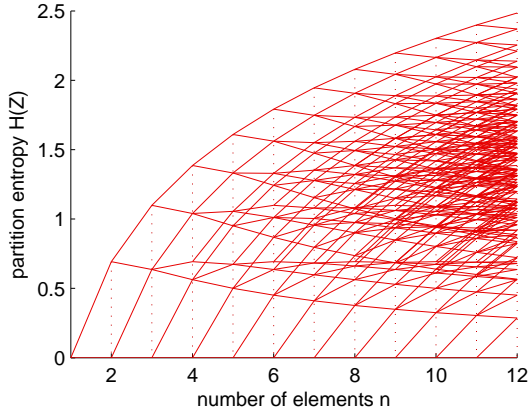
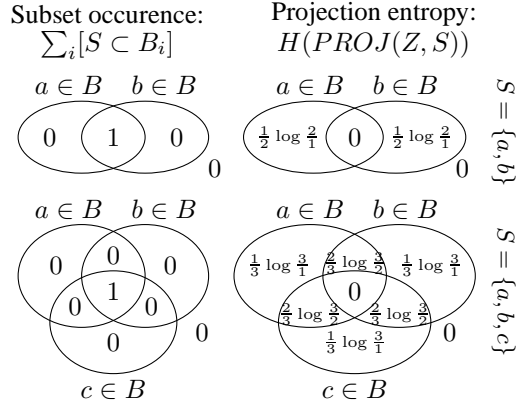

Figure 7: $H(Z)$ in incremental construction of $Z$ — Figure 8: Comparing two subset statistics

We can then compute the per-element information supplied by a partitioning $Z$, by taking a weighted average over its blocks, since each block $B \in Z$ supplies information for a different proportion $|B|/n$ of the elements being partitioned. For large $n$, weighted per-element information reaches its maximum near $|B| \approx n/2$ (Figure 6). Total weighted information for $Z$ gives Shannon's entropy function [17] which can be written in terms of the cumulative statistics (assuming $\phi_{n+1} = 0$):

$$H(Z) = \sum_{i=1}^{|Z|} \frac{|B_i|}{n} \, \mathrm{pei}_n(B_i) = \sum_{i=1}^{|Z|} \frac{|B_i|}{n} \log \frac{n}{|B_i|} = \sum_{k=1}^{n} (\phi_k(Z) - \phi_{k+1}(Z)) \frac{k}{n} \log \frac{n}{k} \quad (11)$$

Entropy of a partitioning increases as its elements become more segmented among themselves. A partitioning with a single block has zero entropy, and a partitioning with $n$ blocks has the maximum entropy $\log n$. Nodes of the tree we examined in the previous section (Figure 3b) were vertically arranged according to their entropies. On the extended tree (Figure 7), $n$th column of nodes represent the possible partitionings of $n$. This tree serves as a 'grid' for both $H(Z)$ and $\phi(Z)$, as they are linearly related with the general coefficient ($\frac{k}{n} \log \frac{n}{k} - \frac{k-1}{n} \log \frac{n}{k-1}$). A similar grid for feature allocations can be generated by inserting nodes for cumulative statistics that do not conserve mass.

To quantify the segmentation of a subset $S$, we compute *projection entropy* $H(PROJ(Z, S))$. To understand this function, we compare it to *subset occurence* in Figure 8. Subset occurence acts as a 'score' that counts the 'successful' blocks that contain all of $S$, whereas projection entropy acts as a 'penalty' that quantifies how much $S$ is being divided and segmented by the given blocks $B \in Z$.

A partitioning $Z$ and a permutation $\sigma$ of its elements induce an *entropy sequence* $(h_1, \ldots, h_n)$ such that $h_i(Z, \sigma) = H(PROJ(Z, S_i))$ where $S_i = \{\sigma_1, \ldots, \sigma_i\}$ for $i \in \{1, \ldots, n\}$. To find subsets of elements that are more closely related, one can seek permutations $\sigma$ that keep the entropies low. The generated subsets $S_i$ will be those that are less segmented by $B \in Z$. For the example problem, the permutation $1, 3, 6, 7, \ldots$ keeps the expected entropies lower, compared to $1, 2, 3, 4, \ldots$ (Figure 4).

## 5 Entropy agglomeration and experimental results

We want to summarize a sample set using the proposed statistics. Permutations that yield lower entropy sequences can be meaningful, but a feasible algorithm can only involve a small subset of the $n!$ permutations. We define *entropy agglomeration* (EA) algorithm, which begins from 1-element subsets, and merges in each iteration the pair of subsets that yield the minimum expected entropy:

**Entropy Agglomeration Algorithm:**

1. Initialize $\Psi \leftarrow \{\{1\}, \{2\}, \ldots, \{n\}\}$.
2. Find the subset pair $\{S_a, S_b\} \subset \Psi$ that minimizes the entropy $\langle H(PROJ(Z, S_a \cup S_b)) \rangle_{\pi(Z)}$.
3. Update $\Psi \leftarrow (\Psi \backslash \{S_a, S_b\}) \cup \{S_a \cup S_b\}$.
4. If $|\Psi| > 1$ then go to 2.
5. Generate the dendrogram of chosen pairs by plotting minimum entropies for every split.

The resulting dendrogram for the example partitionings are shown in Figure 9a. The subsets $\{4, 5\}$ and $\{1, 3, 6\}$ are shown in individual nodes, because their entropies are zero. Besides using this dendrogram as a general summary, one can also generate more specific dendrograms by choosing specific elements or specific parts of the data. For a detailed element-wise analysis, entropy sequences of particular permutations $\sigma$ can be assessed. Entropy Agglomeration is inspired by 'agglomerative clustering', a standard approach in bioinformatics [23]. To summarize partitionings of gene expressions, [14] applied agglomerative clustering by pairwise occurences. Although very useful and informative, such methods remain 'heuristic' because they require a 'linkage criterion' in merging subsets. EA avoids this drawback, since projection entropy is already defined over subsets.

To test the proposed algorithm, we apply it to partitionings sampled from infinite mixture posteriors. In the first three experiments, data is modeled by an infinite mixture of Gaussians, where $\alpha = 0.05, d = 0, p(\theta) = \mathcal{N}(\theta|0, 5)$ and $F(x|\theta) = \mathcal{N}(x|\theta, 0.15)$ (see Equation 1). Samples from the posterior are used to plot the histogram over the number of blocks, pairwise occurences, and the EA dendrogram. Pairwise occurences are ordered according to the EA dendrogram. In the fourth experiment, EA is directly applied on the data. We describe each experiment and make observations:

**1) Synthetic data** (Figure 9b): 30 points on $\mathbb{R}^2$ are arranged in three clusters. Plots are based on 450 partitionings from the posterior. Clearly separating the three clusters, EA also reflects their qualitative differences. The dispersedness of the first cluster is represented by distinguishing 'inner' elements 1, 10, from 'outer' elements 6, 7. This is also seen as shades of gray in pairwise occurences.

**2) Iris flower data** (Figure 9c): This well-known dataset contains 150 points on $\mathbb{R}^4$ from three flower species [24]. Plots are based on 150 partitionings obtained from the posterior. For convenience, small subtrees are shown as single leaves and elements are labeled by their species. All of 50 A points appear in a single leaf, as they are clearly separated from B and C. The dendrogram automatically scales to cover the points that are more uncertain with respect to the distribution.

**3) Galactose data** (Figure 9d): This is a dataset of gene expressions by 820 genes in 20 experimental conditions [25]. First 204 genes are chosen, and first two letters of gene names are used for labels. Plots are based on 250 partitionings from the posterior. 70 RP (ribosomal protein) genes and 12 HX (hexose transport) genes appear in individual leaves. In the large subtree on the top, an 'outer' grouping of 19 genes (circles in data plot) is distinguished from the 'inner' long tail of 68 genes.

**4) IGO** (Figure 9e): This is a dataset of intergovernmental organizations (IGO) [26,v2.1] that contains IGO memberships of 214 countries through the years 1815-2000. In this experiment, we take a different approach and apply EA directly on the dataset interpreted as a sample set of single-block feature allocations, where the blocks are IGO-year tuples and elements are the countries. We take the subset of 138 countries that appear in at least 1000 of the 12856 blocks. With some exceptions, the countries display a general ordering of continents. From the 'outermost' continent to the 'innermost' continent they are: Europe, America-Australia-NZ, Asia, Africa and Middle East.

# 6   Conclusion

In this paper, we developed a novel approach for summarizing sample sets of partitionings and feature allocations. After presenting the problem, we introduced cumulative statistics and cumulative occurence distribution matrices for each of its permutations, to represent a sample set in a systematic manner. We defined per-element information to compute entropy sequences for these permutations. We developed *entropy agglomeration* (EA) algorithm that chooses and visualises a small subset of these entropy sequences. Finally, we experimented with various datasets to demonstrate the method.

Entropy agglomeration is a simple algorithm that does not require much knowledge to implement, but it is conceptually based on the cumulative statistics we have presented. Since we primarily aimed to formulate a useful algorithm, we only made the essential definitions, and several points remain to be elucidated. For instance, cumulative statistics can be investigated with respect to various nonparametric priors. Our definition of per-element information can be developed with respect to information theory and hypothesis testing. Last but not least, algorithms like entropy agglomeration can be designed for summarization tasks concerning various types of combinatorial sample sets.

**Acknowledgements**

We thank Ayça Cankorur, Erkan Karabekmez, Duygu Dikicioğlu and Betül Kırdar from Boğaziçi University Chemical Engineering for introducing us to this problem by very helpful discussions. This work was funded by TÜBİTAK (110E292) and BAP (6882-12A01D5).

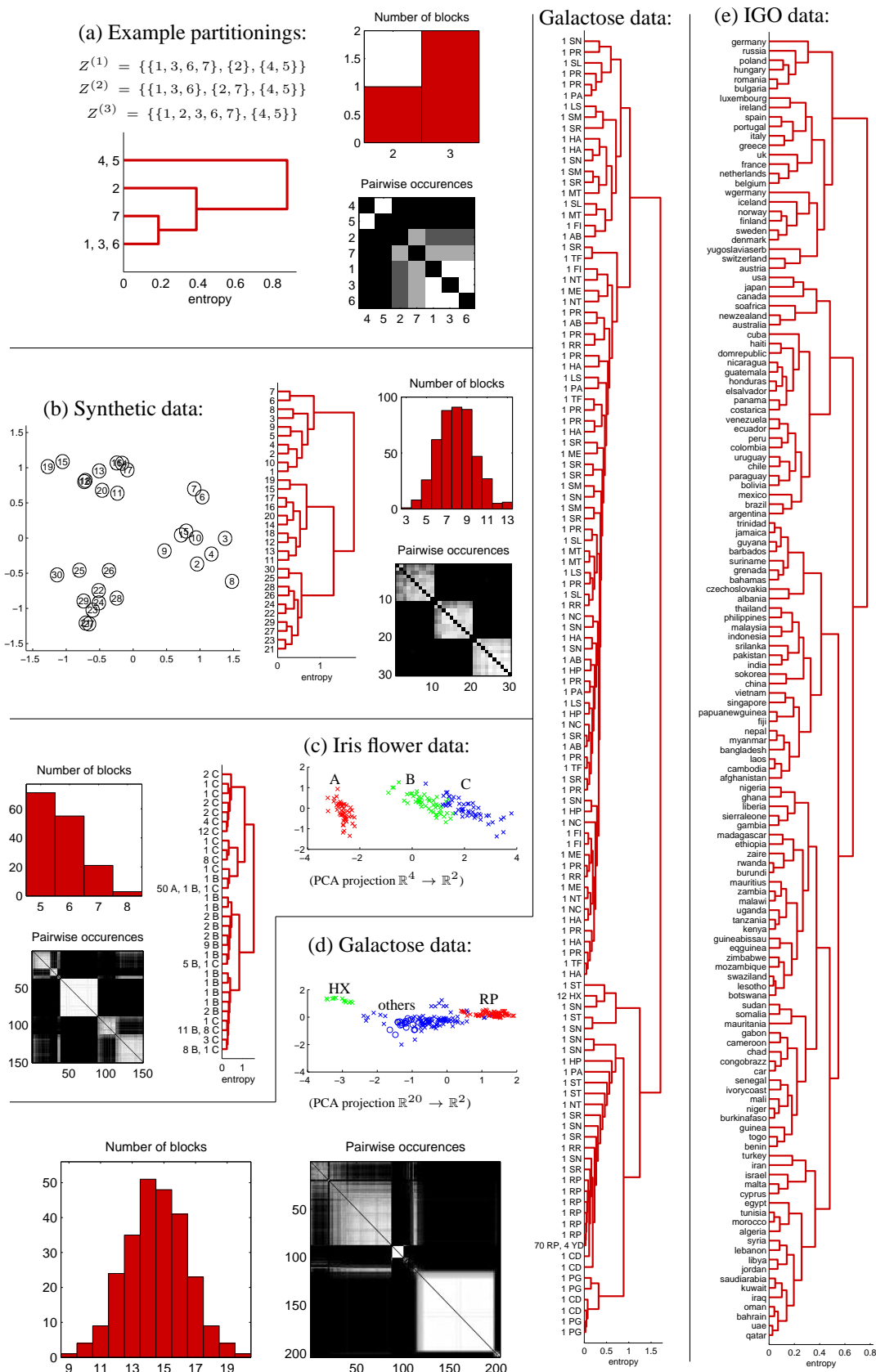

Figure 9: Entropy agglomeration and other results from the experiments (See the text)

## Footnotes

[1] In methods such as collapsed Gibbs sampling, slice sampling, retrospective sampling, truncation methods

[2]We use the term 'partitioning' to indicate a 'set partition' as distinguished from an integer 'partition'.

[3]The symbol '$\vdash$' is usually used for integer partitions, but here we use it for partitionings (=set partitions).

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
