[Reviews · NeurIPS 2013]

Submitted by Assigned_Reviewer_5

The authors propose 2 statistics for summarizing samples from the prior or posterior distribution of random partitions and feature allocations and an algorithm which is a variation of agglomerative clustering with a different merging rule in terms of one of their proposed statistics.

Their first proposal is what they call a COD matrix, this is constructed from a Young diagram of a partition and a projection operation which helps distinguish more segmented clusters from less segmented ones. Their second proposal is to compute the entropy of this projection rather than build its COD matrix. They show some experiments for various data sets in which they use a DP mixture model and show the corresponding posterior distribution for the number of clusters, co-clustering probabilities and dendrograms obtained from their entropy agglomeration algorithm.


They motivate the problem really well and it is quite an important open question but it is not clear if their proposal is a solution to it. This is because they use the samples of the posterior distribution as an input for their agglomerative clustering algorithm and show the corresponding dendrograms, which they claim to be useful summary statistics of the posterior distribution over the number of clusters. Usually, dendrograms are seen as summaries of any agglomerative clustering algorithm so it is not really clear why they can also be interpreted as summary statistics for the posterior distribution of number of clusters. Furthermore, they don't elaborate on this point or compare it to Medvedovic & Sivaganesan (2002) agglomerative clustering algorithm which also uses samples from the posterior distribution for computing the co-clustering probabilities. For example: What does the tree depth tells us about the posterior distribution?, Does it recovers the number of clusters in a better way? Which characteristics of the posterior are reflected on the dendrogram? Is it sensitive to different priors? ( a simple case would be to compare CRP v.s. 2-CRP) I think if we had this comparisons and convincingly show that their algorithm is better and that the summary statistics reflect novel/specific information about the posterior it would be really interesting. Alternatively, if the algorithm performs well against any of the existing AC ones then it would be good to see its performance in various settings and detailed explanation of its properties. The latter would be useful in its own right whether or not it is related to summarizing posterior distribution's characteristics.

A few specific comments:

Line 19: It is correct to say that the MAP might not be good in situations where the posterior is diffuse, I guess you mean uniform-like? It might also be multimodal, skewed (mode and mean differ), high variance,for example, so the MAP is not necessarily a good choice although it depends on the (underlying) loss function.

Line 52: Since a Dirichlet Process is a discrete random probability measure, when sampling from it it induces a random partition (the ties will belong to the same cluster). If you sample, say, n times from the DP, then you obtain a draw from a random partition of size k<=n with its corresponding frequencies n_i, i=1,...k. The joint distribution of k and the frequencies is what people call a CRP, but this is just a joint distribution of this objects so there is no need to 'generalize' entropy since it is defined for any probability distribution. Furthermore, it is incorrect to say that partitions are interpreted as probability distributions for the above reasons.
Line 65: I don't think you need to make the footnote's distinction since an integer partition is just a set partition where your total set is {1,...,n}.
Line 70: It is not clear what k is.
Line 72: Reference [3] is not the appropriate one for this, there should be an earlier one.
Line 74: Usually, the term 'collapsed Gibbs sampler' in infinite mixture models case, refers to the fact that, rather than having to sample a parameter for each observation (n operations in a dataset of size n) you only need to sample a parameter for each cluster (k <= n operations in a sample of size n). It's more precise to call this ' conjugate marginal sampler' since we can effectively integrate out parameters due to the conjugacy of the base distribution with the likelihood.
Line 77: Reference [7] is not the appropriate one for equation (2) since this is Algorithm 3 (conjugate case and Neal's paper extends this for the non-conjugate case in Algorithm 8). The correct reference is MacEachern (1994) and also Neal (1992) (Check the references in Neal's paper).
Line 136: Might be good to reference Z. G. Su (2007) "Asymptotic analysis of random partitions", for example, for the Young diagram part.
Line 154: I don't understand what you mean by 'conjugacy' here. Young diagrams are a graphical representation of the cumulative statistic by definition, see above reference.
Line 162: I think it is incorrect to say that 'the cumulative statistic is always the conjugate partition on the integer partition of Z'. The definition of a conjugate partition is: 'the conjugate partition of a partition is the partition whose Young diagram is the transpose of the Young diagram of the first one'. Check reference above/explain why in this case it's equivalent.
Line 191: Indeed, since a CRP is projective and exchangeable then the distribution of the random partition doesn't depend on the particular permutation but this does not explain why the argument inside the expectation doesn't depend on the permutation since the incremental construction clearly does. So it would be nice to see why the resulting COD matrix is independent of sigma.
Line 215: This legend is misleading since the COD matrices here correspond to {1,2,..,7} and Z(1) of the previous page, Line 144. Furthermore, it is not clear why these 3 cases are illustrative or just randomly picked.
Line 229: The incremental formulation is interesting but it's not used in any other part of the paper, neither the COD matrix.
Line 239: It is important to make the distinction between CRP and 2-CRP.
Line 252: Is n the number of data points? It might help to clarify this.
Line 300: It might help to give a toy example with feature allocations to get a better grasp the definition of entropy of equation (11). Also, explain why the cumulative statistics for feature allocations do not preserve mass.
Line 307: The projection operation was nicely motivated previously in the toy example (Line 145) since it helped to distinguish between more segmented partitions. However, it is not immediately obvious what does it mean to take the entropy of this.
Line 310: Figure 4 should be around here rather than in the previous section.
Line 321: It might help to explain how the posterior samples are used in the pseudocode,e.g. rather than calculating the expected entropy you use the samples from the posterior distribution to estimate it, assuming that this is indeed how you use the posterior samples.
Line 327: It is not clear for me how does the dendrogram encode useful information about the posterior distribution of the partition. It would be nice to elaborate about this, e.g. was does the tree depth tell us.
Line 332: It is not immediately obvious why this method is principled rather than a heuristic. Furthermore, it would be useful to provide more arguments of why is it better than any of the existing agglomerative clustering approaches or at least against Medvedovic & Sivaganesan (2002) e.g. does it recover the correct number of clusters better.

Quality: this is a well written paper.

Clarity: the importance of the problem is clearly explained and the paper has nice figures to illustrate the concepts but there are parts which are not well connected.

Originality and Significance : the paper introduces nice ideas however I am still not sure if it is indeed a solution to the problem of summarizing samples from a random partition.


Summary: The authors propose 2 statistics for summarizing samples from the prior or posterior distribution of random partitions and feature allocations and an algorithm which is a variation of agglomerative clustering with a different merging rule in terms of one of the proposed statistics. It is well written but might need more comparisons and/or explanations.

Submitted by Assigned_Reviewer_6

Summary:
The authors propose novel approaches for summarizing the posterior of partitions in infinite mixture models. Often in applications, the posterior of the partition is quite diffuse; thus, the default MAP estimate is unsatisfactory. The proposed approach is based on the cumulative block sizes, which counts the number of clusters of size ≥ k, for k=1, …,n. They also examine the projected cumulative block sizes, when the partition is projected onto a subset of {1,...,n}. These quantities are summarized by the cumulative occurrence distribution, the per element information of a set, the entropy, the projected entropy, and the subset occurrence. Finally, they propose using an agglomerative clustering algorithm where the projection entropy is used to measure distances between sets. In illustrations, the posterior of the partition is summarized by the dendrogram produced from the entropy agglomerative algorithm, along with existing summaries such as the posterior histogram of the number of clusters and the pairwise occurrences.

Strengths:
The authors develop a new method of summarizing the posterior of the partition with a discussion and analysis of many of the quantities involved.
An algorithm is developed to summarize the partition based on the summary statistics introduced.
Demonstrations of the algorithms on many datasets are included.

Weaknesses:
The extension to feature allocation is quite underdeveloped and it seems misleading to have it in the title. Also, the discussion on feature allocation in its current format seems to disrupt the flow of the paper. Possibly, it would be best to leave the full extension to feature allocation for later work, briefly mentioning it in the conclusion, or to create a small separate section on feature allocation towards the end of the paper.
The algorithm developed feels a bit heuristic, as it is not developed from decision theory. There are alternative methods for summarizing the partition based on decision theory, such as Quintana and Iglesias (2003), Lau and Green (2007), and Fritsch and Ickstadt (2009). These papers are not cited or discussed.
The new summary tool presented in the applications, i.e. the dendrogram from the entropy agglomerative algorithm, is not so easy to interpret, and it is not clear how much it improves our understanding of the partition over the other existing summary tools.
pg.1 line 41, also slice sampling, retrospective sampling, and truncation methods, give posterior samples of the partition.
pg. 3 line 108, block sizes are not necessarily better summary statistics than pairwise occurrences; in particular, if you know whether or not each pair of data points are clustered together than you know the partition structure (up to a relabeling of the clusters), but if you know all block sizes, this does not determine the partition structure. Thus, one could argue that the pairwise occurrences are better at “capturing the partition structure”.
pg. 3 line 126, I believe PROJ(Z^(2), S_3) is incorrect.
pg. 3 line 138, a more appropriate shorted name could be cumulative block sizes.
pg. 3 line 138, It is a bit misleading to say the pairwise occurrences can be written in terms of the cumulative block sizes \phi(Z). In fact, the correct statement would be that pairwise occurrences can be written in terms of the projected cumulative block sizes \phi( PROJ(Z,S)) for every set S of possible pairs of data points.
pg. 3 line 140, should say “probability mass function” not “probability distribution”.
In the illustrations, it would be nice to include a comparison of the entropy agglomerative algorithm with agglomerative algorithm based on the pairwise occurrences of Medvedovic and Sivaganesan (2002).
The illustrations are presented based on a relatively small number of posterior samples of the partition. Can the algorithm handle a larger number of partitions?

Quality: This is a technically sound paper with theoretical developments and experimental results.

Clarity: This is a clear paper with several figures to depict the underlying concepts introduced. However, it is a bit confusing that after introducing the various concepts and statistics, the only new summary tool presented in the illustrations is the dendrogram from the entropy agglomerative algorithm. Furthermore, the inclusion of feature allocation is underdeveloped and disrupts the flow of the paper.

Originality: Novel summary statistics are introduced as well as a novel algorithm and dendrogram plot to summarize the posterior of the partition.

Significance: I believe these results are important and do offer a new way of summarizing the partition. However, it feels a bit ad-hoc, and I would like to see the motivation of this approach over the more elegant summaries of Lau and Green and Fritsch and Ickstadt based on decision theory. Moreover, I am not convinced that the EA dendrogram provides much new insight for the partition structure over the pairwise occurrences. In particular, it would be interesting to see a comparison of the EA dendrogram vs. the pairwise occurrences dendrogram of Medvedovic and Sivaganesan (2002).
Summary: The authors address the important and interesting problem of how to summarize the posterior of the partition and do so by introducing novel summary statistics and developing an algorithm based on these summaries. While it is a technically sound paper with several nice illustrations, I am concerned about the improvement of the proposed summaries over existing summaries including those not mentioned in the text Lau and Green (2007) and Fritsch and Ickstadt (2009).

Submitted by Assigned_Reviewer_7

The paper presents a principled method for summarizing a set of sampled cluster partitionings. The authors provide a theoretical analysis that describes how cumulative statistics of cluster sizes can be used to compare partitionings, and makes connections between these statistics and a notion of entropy for partitionings. They further present a simple greedy algorithm that can be used to construct a hierarchical clustering (dendrogram) of data items given a set of sampled partitionings that minimizes entropy at each split. Experiments on several datasets show that the dendrograms can usefully aid human interpretation of relationships between data items.

Quality: The paper treats its subject thoroughly and insightfully.

Clarity: The paper is well-written and easy-to-follow, especially the running simple examples used to illustrate each new idea. The toughest parts to understand were some of the figures (6 and 7), mostly because captions were spartan. Some experimental details were also too terse, especially the one on feature allocation.

Originality: Definitely novel material. The connection between cumulative statistics of block sizes and notions of entropy is new (at least to me), and could open several lines of investigation.

Significance: The presented EA algorithm is both elegant and simple-to-implement, so I expect it could be widely used by the community. Moving beyond flat clusterings to offer more rich interpretations of data separability is a useful line of research. Popular existing approaches rely on heuristic linkage criteria, so this approach could be a nice alternative.

I am sympathetic to arguments that the proposed approach to building dendrograms is principled and should be preferred to conventional agglomerative techniques. However, a natural question is whether there are noticeable differences between the proposed EA and more traditional linkage criteria. I understand this isn't the main focus of the paper (and that both methods don't solve exactly the same problem), but this will be what many folks want to know in practice so any thoughts are appreciated.

How sensitive is the method to samples that aren't quite "independent"? One-site-at-a-time Gibbs samplers are notorious for mixing poorly on datasets of even moderate scale. Adding some discussion on this point would be appreciated. I worry that for most large-scale datasets of interest, obtaining a large-set of truly independent samples from the posterior is arduous, which limits the applicability of the method.

One question for the authors is whether the EA algorithm optimizes (or attempts to optimize) some global objective function about the total entropy of a dendrogram. The algorithm is greedy in nature, so a natural question is what might be lost (if anything) by always taking the best local choice.

Summary: A method for summarizing sampled cluster partitions that has interesting theoretical connections between entropy and cumulative statistics, and leads to an elegant, simple algorithm for hierarchical clustering. I have some concerns about whether the dendrograms will be preferred to agglomerative methods in practice, and how well the method scales to large data (where obtaining "independent" posterior partitionings is trickier), but overall a welcome paper at NIPS.
Author Feedback

Author rebuttal: We are sincerely grateful to the reviewers for their constructive comments and for their time in creating these detailed and high quality reviews. We will alter the manuscript accordingly.

As the reviewers affirm, developing summary statistics for distributions over partitions is important in applied work but a difficult question even to formulate properly. Considering the surge of interest in sampling methods for nonparametric models, the related literature for characterization of empirical distributions (such as samples from a MCMC run) is surprisingly sparse. Clearly, we are not claiming that we provide the final answer. Our humble contribution is providing conceptual tools (COD matrix and entropy) to allow a clear and coherent formulation of this question.

Here, we propose entropy agglomeration (EA) as a practical tool for investigating concrete problems in terms of the concepts we present. We are unable to claim the superiority of EA to any previous methodology since its basic principles have only been defined and demonstrated in the current paper.

Since all three reviewers recommended including a comparison with Medvedovic & Sivaganesan 2002, we would like to explain why we were hesitant to do so. M&S 2002 is a very important paper in pointing out the basic problem to 'summarize' a sample set of partitionings (or 'clusterings'). However, as the authors note in their follow-up work, they do not propose a method that addresses this problem but 'heuristic modifications that effectively circumvent' it (Liu,Sivaganesan,Yeung,Guo,Bumgarner,Medvedovic-Bioinformatics,2006 and Medvedovic,Yeung,Bumgarner-Bioinformatics,2004).

To compare M&S to our approach, please note that splits in the proposed EA dendrogram plot reflect entropy values that quantify 'segmentedness' among elements (as we define it in the paper), whereas splits in an M&S dendrogram plot reflect *averages over pairwise probabilities that belong to different pairs of elements*, a quantity that does not mean much. For this reason, we only included pairwise occurrence matrices and not the corresponding dendrograms.

Realising the variety of questions and concerns that our paper calls forth, and considering the page limits, we would like to formulate and address these additional questions in future work, as we indicate in the paper.

We wish to provide some comments and specific answers to the raised concerns. To keep the text short, we do not enumerate our corrections.

R5

- Both COD matrix and entropy are representations for the general (non-exchangable) case.

- We prefer to say 'number of blocks' instead of 'number of clusters', since blocks directly belong to partitionings, whereas 'cluster' may have different meanings depending on the context.

- Sensitivity of entropy estimates to DP and PDP priors were investigated by Nemenman et al (2002) and Archer et al (2013).

- We use samples from infinite mixture posteriors in examples, but our statistics can handle any sample set of partitionings/feature allocations.

L19: Concepts like 'multimodality, skewedness' are more useful on continuous parameter spaces and associated densities, but they may be misleading in discrete (combinatorial) spaces. We prefer to use 'diffuse' similar to its use in Bayesian modelling such as 'a noninformative diffuse prior'. In Section 4 we define 'entropy' to measure 'segmentedness', a concept like diffuseness for combinatorial spaces.

L52: We agree that there is no need to 'generalize entropy' and 'interpret partitions as probability distributions'. These expressions refer to Simonvici (2007). We take a different approach by re-formulating entropy in terms of per-element information.

L65: We distinguish set partitions from integer partitions since we project them onto subsets. Projection is undefined for integer partitions, since they consist of integers.

L162: In Figure 1a 'cumulative statistic' and 'integer partition of Z' correspond to the same Young diagram vertically and horizontally. Since the Young diagram of one vector is the transpose of that of the other, they must be conjugate partitions.

L191: We only say that 'expected COD matrix' of CRP is 'independent of sigma', COD matrix inside the expectation is not independent of sigma.

L229: COD matrix and entropy are 'used' on relatively simple examples to better demonstrate their novelty and descriptive power.

L327: We would like to investigate tree-depth and other properties of EA in future work.

R6

- We agree that feature allocations need more elaboration, but we think it is helpful not to separate them from partitionings in formulating the problem: partitionings (as special cases) are helpful in making formulations but these formulations aim for feature allocations (general case).

- We understand summarization as 'extracting information'. While we admit that viewing the problem from a decision theory perspective is certainly a very viable suggestion, we did not directly relate our approach to decision theory since it is not clear to us if extracting information from a sample set of partitionings would necessarily require decisions that incur loss.

- For T partitionings of n elements, EA runs for n-1 iterations. In the ith iteration, it uses entropies of pairs from n-i+1 subsets. Each entropy computation loops through the T partitionings, projecting them onto a subset. It is a relatively quick algorithm but can be further optimized for speed.

R7

- If the Gibbs sampler has not converged, the obtained samples are from a different density, but this issue, whilst very important, is rather tangential to the problem we try to address here. Instead of samples from a Gibbs run, EA can be also applied directly to data (e.g., see the IGO dataset).

- Is there a global function that EA effectively optimizes? This is one of the important questions that we would like to investigate in future work.